# The Function of KDEL Receptors as UPR Genes in Disease

**DOI:** 10.3390/ijms22115436

**Published:** 2021-05-21

**Authors:** Emily S. Wires, Kathleen A. Trychta, Lacey M. Kennedy, Brandon K. Harvey

**Affiliations:** Molecular Mechanisms of Cellular Stress and Inflammation, Intramural Research Program, National Institute on Drug Abuse, Baltimore, MD 21224, USA; kathleen.trychta@nih.gov (K.A.T.); lacey.kennedy@nih.gov (L.M.K.)

**Keywords:** KDEL receptor, endoplasmic reticulum, unfolded protein response, ER resident proteins, disease, exodosis

## Abstract

The KDEL receptor retrieval pathway is essential for maintaining resident proteins in the endoplasmic reticulum (ER) lumen. ER resident proteins serve a variety of functions, including protein folding and maturation. Perturbations to the lumenal ER microenvironment, such as calcium depletion, can cause protein misfolding and activation of the unfolded protein response (UPR). Additionally, ER resident proteins are secreted from the cell by overwhelming the KDEL receptor retrieval pathway. Recent data show that KDEL receptors are also activated during the UPR through the IRE1/XBP1 signaling pathway as an adaptive response to cellular stress set forth to reduce the loss of ER resident proteins. This review will discuss the emerging connection between UPR activation and KDEL receptors as it pertains to ER proteostasis and disease states.

## 1. Introduction

The endoplasmic reticulum (ER) is a dynamic intracellular organelle integral to cellular homeostasis. Although the ER plays critical roles in lipid synthesis [1], calcium storage [2], carbohydrate metabolism [3], and xenobiotic detoxification [4], it is most commonly known for its role in protein synthesis [5], protein folding [5], protein modifications via post-translational modifications [5], and initiating protein degradation processes [6]. The extensive quality controls of the ER ensure that properly folded, mature proteins reach their destinations inside and outside of the cell. The unfolded protein response (UPR) is an adaptive quality control mechanism employed by cells in response to ER proteostasis disruptions. The UPR is comprised of three distinct arms, IRE1 (inositol-requiring kinase 1), PERK (double-stranded RNA-activated protein kinase (PKR)-like ER kinase), and ATF6 (activating transcription factor 6), each activating signal transduction pathways to alleviate the burden of misfolded proteins [7]. Another pathway that helps to maintain proteostasis in the ER is the KDEL receptor retrieval pathway responsible for retaining ER resident proteins (e.g., chaperones and isomerases) many of which are involved in the initiation and propagation of the UPR. In this review, we will discuss recent evidence supporting that KDEL receptors are UPR genes used to restore proteostasis under conditions of ER stress, especially stress related to ER calcium depletion.

## 2. KDEL Retrieval Pathway and KDEL Receptors

Many non-membrane proteins translated into the ER are transient occupants of the ER lumen. These proteins are folded and modified in the ER before moving on to the Golgi for further modification and trafficking to their destination. However, there are also proteins such as chaperones and isomerases that reside in the ER lumen to assist in the folding and maturation of nascent proteins. Other ER resident proteins perform a variety of functions including calcium binding, acting as cellular stress sensors, and catalyzing biochemical reactions (i.e., esterases and peptidases) [8]. Under homeostatic conditions, proteins are maintained in the ER lumen via a carboxy-terminal ER retention sequence (ERS). The first ERS identified was a four amino acid sequence Lys-Asp-Glu-Leu (KDEL) found on three ER localized proteins (BiP/Grp78, Grp94, and protein disulfide isomerase) [9]. The KDEL sequence was shown to be critical for localizing proteins to the endoplasmic reticulum by interacting with a receptor, the KDEL receptor, located in the Golgi [10,11,12]. This interaction is thought to elicit a receptor conformational change and complex with COP-I vesicles for permitting retrograde transport from the Golgi to ER, where the ERS-containing protein is released from the KDEL receptor [13]. This protein retrieval pathway is now known to work on many carboxy terminal peptides beyond the canonical KDEL sequence [8,14,15,16].

The KDEL receptor has three highly homologous isoforms, KDELR1, KDELR2, and KDELR3, encoded by three genes, *ERD2.1*, *ERD2.2*, and *ERD2.3*, respectively [17]. KDEL receptors are a seven-transmembrane protein that resembles G-protein coupled receptors (GPCR) based on topology and transmembrane helices, but they are considered members of the PQ-loop protein family [18,19]. The isoforms are thought to be ubiquitously expressed among tissue types, although the relative mRNA expression of each isoform varies among cell lines and rodent tissue type, suggesting a tissue- or cell-specific expression [8,14]. A bias of the KDEL receptor isoforms for specific ERS tails has been postulated, but the isoform-specific functions are unclear [8,14]. It is plausible that relative expression of KDELRs may be related to the composition of ERS-containing proteins expressed in the same cell, but further study is needed.

KDEL receptor localization is not solely limited to the ER–Golgi intermediate compartment (ERGIC). The presence of KDEL receptors at the cell surface has been reported, suggesting a multifaceted role for KDEL receptors. Mesencephalic astrocyte-derived neurotrophic factor (MANF) is an ER stress-responsive protein with an ERS [20]. Henderson et al., 2013, demonstrated that ER calcium depletion increased the cell surface KDEL receptor expression and indicated MANF interaction with KDEL receptors at the cell surface [21]. This suggests that alterations to homeostatic conditions, as is often observed in pathological diseases, can change the expression and localization of KDEL receptors. ERS proteins, such as protein disulfide isomerase (PDIs) and cerebral dopamine neurotrophic factor (CDNF), have been shown to evade the canonical Golgi–ER retrograde transport during ER calcium depletion, reaching the extracellular space [8,22]. Cell surface KDEL receptors have been postulated as a putative receptor for extracellular CDNF, suggesting that KDEL receptors adaptively re-distribute in response to changes to the ER microenvironment [22]. Moreover, KDEL receptors have been implicated in the trafficking of ERS proteins to the cell surface [23]. Interestingly, cell-surface clustering of KDEL receptors is dependent on cargo, dose, and temperature [24]. It is therefore plausible to speculate that increased ER stress and UPR activation can overwhelm the ERGIC KDEL receptor pathway, increase the flow of KDEL or KDEL-like cargo along the secretory pathway, and lead to the trafficking of both cargo and KDEL receptors to the cell surface.

The KDEL retrieval pathway, or ERS retrieval pathway, is sensitive to changes in the ER calcium. Specifically, the depletion of ER calcium elicits the secretion of ERS proteins from the cell by overwhelming the ERS retrieval pathway [8]. This mass departure of ER resident proteins in response to ER calcium depletion is referred to as “exodosis”. Overexpressing KDEL receptors attenuates exodosis, whereas knocking down KDEL receptors potentiates exodosis [8]. KDEL receptor mRNA expression is also increased in response to ER stress, including that caused by ER calcium depletion [8]. Collectively, these data suggest that KDEL receptors are UPR genes that are part of an adaptive response to counter the loss of ERS proteins, including some UPR targets that are being upregulated in response to ER stress.

## 3. KDEL Receptors as UPR-Regulated Genes

Recently, KDEL receptors, specifically *KDELR2* and *KDELR3*, were shown to be upregulated in a variety of cell types following treatment with two commonly used chemical inducers of the UPR, tunicamycin and thapsigargin [8,25]. Samy et al., 2020, observed moderate increases in *KDELR1*, *KDELR2*, and *KDELR3* mRNA expression following tunicamycin treatment, while other well-known UPR downstream targets, PDI (*P4HB*), BiP (*GRP78*, *HSP70,* and *HSPA5*), calreticulin (*CALR3, CRT2*), and Grp94 (*HSP90B1, GRP94, TRA1*) had more pronounced levels of upregulation. Alternatively, Trychta et al., 2018, highlights putative XBP1 binding sites in *KDELR* genes, with *KDELR2* and *KDELR3* having more sites than *KDELR1*, correlating with an increased mRNA expression of *KDELR2* and *KDELR3* in response to ER stress. These data suggest that *KDELR2* and *KDELR3* are downstream targets of the IRE1 pathway [8]. The IRE1 pathway was originally identified in yeast and is the most evolutionarily conserved arm of the UPR [26]. BiP is an abundant ER chaperone protein involved in the recruitment of misfolded proteins for refolding or degradation, and acts as the primary molecular sensor of the UPR [27,28]. Dissociation from BiP permits IRE1 oligomerization, leading to auto-phosphorylation and promotion of ribonuclease intron splicing of *XBP1*1 mRNA. Once translated, XBP1 is a transcription factor capable of upregulating UPR-associated genes such as ER chaperones for protein folding quality control, and degradation-associated proteins for the removal of aberrant proteins [8,29,30]. To explore whether the high prevalence of putative XBP1 binding sites in *KDELR2* and *KDELR3* genes implicated the IRE1 pathway in the regulation of their expression during UPR, cells were treated with an IRE1α kinase inhibitor that attenuated the thapsigargin-induced expression of *KDELR2* and *KDELR3*. Previous work using a Tet-inducible spliced *XBP1* reported an approximate three-fold increase in *KDELR3* [31]. Using the same approach, *KDELR2* and *KDELR3*, but not *KDELR1,* were shown to be upregulated in a doxycycline-dependent manner, further supporting that *KDELR2* and *KDELR3* are UPR genes activated by the IRE1 pathway [8]. While these observations support that KDEL receptors are UPR-regulated genes, it is prudent to consider variations of regulation among cell types, secretion profiles, and UPR activation susceptibility. To this consideration, *KDELR1*, *KDELR2*, and *KDELR3* displayed similar upregulation in CHO cells following tunicamycin treatment [25], supporting the notion that KDEL receptor activation may be cell type-specific and/or stressor-dependent. The remainder of this review will explore the role of KDEL receptors and UPR activation in various pathological sates. Examining KDEL receptor isoforms as UPR-regulated genes may provide insight into possible therapeutic targets and their role in retaining other UPR targets that have an ERS tail (e.g., BiP/Grp78).

## 4. KDEL Receptors, UPR, and Disease

The UPR is responsible for maintaining cellular proteostasis following ER stress associated with diseases. The differential transcriptional programs activated by the three prongs of the UPR allow for a variety of cellular responses that may be tailored to the nature of the ER stress stimulus. The complex etiology of most diseases and the potential flexibility of the UPR may allow cells to fine tune a UPR response to a select pathology. As a transcriptional target of the IRE1/XBP1 pathway of the UPR, KDEL receptors represent genes of interest for modulating ER stress caused by disease.

### 4.1. KDEL Receptor and Autophagy

Early studies of KDEL receptors demonstrated that the interaction of a KDEL receptor ligand leads to the activation of the p38 MAP kinase pathway, which was previously shown to be upregulated in response to ER stress [32]. Giannotta et al., 2012, reported that KDEL receptors colocalize with G_s_ and G_q_/11 GPCRs on the Golgi. The delivery of a KDEL-containing protein leads to G_q_/11-dependent activation of the Golgi pool of the Src Family Kinases (SFK). This SFK signaling is required for the transport of cargo from the Golgi to the plasma membrane and is KDEL ligand-dependent [33,34]. The signaling of KDEL receptors via G_s_ GPCRs can modulate autophagy [35,36]. Autophagy is essential for the degradation of misfolded proteins and the UPR has been shown to induce autophagy through several pathways [37,38]. Wang et al., 2011, showed that KDEL receptors localize to the lysosomes when overexpressed, which correlated with an increase in autophagy. The rise in autophagic activity was thought to be through an ERK/MEK dependent signaling pathway because there was an increase in phosphorylated MEK and ERK after KDEL receptor overexpression [35]. Tapia et al., 2019, expanded on these observations by showing that the stimulation of KDELR1 with a KDEL ligand led to the relocation of lysosomes to the perinuclear area, in a protein kinase A (PKA) dependent manner. KDELR1 activates the G_s_/PKA pathway, which proceeds to phosphorylate dynamin light chain roadblock 1 (DynLRB1) and leads to the relocation of lysosomes to the Golgi. DynLRB1 phosphorylation and consequent lysosome relocation facilitated the exit of an exocytic transport reporter to the plasma membrane through the autophagy mediated degradation of lipid droplets [36].

There is growing evidence that autophagy is both triggered by the UPR and regulates ER function. For example, pharmacological inducers of the UPR, such as thapsigargin and tunicamycin, are known to drive autophagosome formation, specifically through IRE1 signaling [39]. Autophagosomes are now thought to play a role is shaping the ER during stress through a process called ER-phagy [40]. Given that KDEL receptor expression is increased in response to the UPR, in addition to their role in stimulating autophagy, these receptors might represent a missing link between the two highly interconnected processes.

### 4.2. Neurodegeneration

The UPR has been associated with many neurodegenerative diseases, and more detailed information on these associations can be found in reviews such as Hetz and Saxena, 2017, and Xiang et al., 2017 [41,42]. For the purposes of this section, we will discuss the role of the IRE1 pathway, ERS protein BiP, and autophagy in neurodegeneration, each of which relates to the KDEL receptors and has been implicated in neurodegeneration [42,43,44]. The IRE1 pathway, specifically, is implicated in Parkinson’s disease, Huntington’s disease, and amyotrophic lateral sclerosis (ALS) [42]. In a Parkinson’s model, developmental ablation of *XBP1* in the mouse substantia nigra pars compacta was found to be protective against 6-OHDA induced neurodegeneration. However, downregulation of *XBP1* in the same area in adult mice led to increased neurodegeneration, suggesting a dynamic or aging related role of XBP1 in the nervous system [45]. Likewise, studies investigating the IRE1 pathway in Huntington’s disease and ALS found that XBP1 deficiency was protective against neuron degeneration in their respective models, and also lead to an increase in autophagy dependent clearance of Htt and SOD-1 [46]. The dynamic role that XBP1 seems to be playing in different neurological states suggests IRE1 activation has both negative and positive effects in the context of neurodegeneration, depending on the downstream effectors involved. Given the data that KDEL receptors are potential targets of the IRE1 pathway, it would be beneficial to examine isoform specific expression of KDEL receptors in the neurodegeneration models mentioned above to further investigate what role KDEL receptors play in the context of these developmental and pathological states.

KDEL receptors as part of the IRE1 pathway are only one potential connection of these proteins to neurodegenerative diseases. A study specifically looking at the relationship between KDEL receptors and the ER localized protein, BiP, showed that deleting the carboxy terminal KDEL tail from BIP in mice led to severe motor disabilities, as well as an accumulation of misfolded proteins [43]. It has been suggested that defective BiP might also lead to impairments in KDEL receptor function, as a result of a lack of KDEL receptor downstream signaling [47]. While a more recent study demonstrated that other ER retention sequences are sufficient to facilitate retention, the role of the KDEL receptor mediated the retention of ER resident proteins, and what effect the dysregulation of this process might have on disease states has not been fully explored [8].

The KDEL receptor’s function in ER protein retention is not the only role that might have neurodegenerative disease implications. KDEL receptors can also modulate autophagy which is impaired in some neurodegenerative diseases [44]. A study demonstrated that misfolded proteins, such as huntington (Htt) in Huntington’s disease, superoxide dismutase 1 (SOD1) in ALS, and α-synuclein (a-syn) in Parkinson’s disease, were all shown to increase KDEL receptor expression, and the knock down of KDEL receptors led to increased levels of the disease-related proteins. These reciprocal effects between KDEL receptors and misfolded proteins were lost in autophagy deficient cell lines, suggesting the actions of KDEL receptors to clear misfolded proteins is autophagy-dependent [35]. The KDEL receptor isoform used in the aforementioned study was not specified, so additional exploration into the relationship between KDEL receptor isoforms and pathogenic protein accumulation would be informative. Various neurogenerative disease studies have now implicated the UPR and KDEL receptor mediated functions in their pathologies. The link between UPR activation and KDEL receptor regulation warrants further investigation in t neurodegenerative diseases, to further understand the connection between these processes in pathological states.

### 4.3. Ischemia

Ischemia induces ER stress and triggers the activation of the UPR. Following cerebral ischemia, UPR target genes and proteins, including Erp72, CHOP, and BiP, are upregulated in the hippocampus [48,49,50]. Similarly, in myocardial ischemia, BiP, ATF6, CHOP, and spliced XBP1 protein and transcript levels are upregulated [51]. In both cerebral and myocardial ischemia, KDEL receptors are also upregulated, perhaps to restore cellular functioning (Table 1 [52,53,54]). Following an ischemic assault, maintaining protective ER chaperones in the ER lumen and promoting angiogenesis are linked to better outcomes and are associated with the induction of the IRE1/XBP1 prong of the UPR [55]. An upregulation of KDEL receptors could be important for promoting these protective programs by ensuring that the ER proteins needed for protein folding, trafficking, and maturation are present within the ER. Additionally, one study proposed that the cardiomyokine CDNF must interact with KDEL receptors to exert its cardioprotective effects, and this same principle could apply to other ERS-containing proteins as well [22]. This suggests that KDEL receptor upregulation may benefit ischemic cells through a role in signaling pathways (i.e., autophagy) and reduce the loss of ER resident proteins. Autophagy, in particular, is thought to have a dual role in ischemia, with detrimental effects being seen during ischemia, but beneficial effects seen during reperfusion. It is possible the KDEL receptors may be involved in this ischemia-induced autophagy, and it is worth noting that while KDEL receptors are upregulated following myocardial infarction in pigs, they are downregulated in a model of ischemic postconditioning [53]. KDEL receptors may have a multi-faceted role in ischemia and their regulation needs to be examined more closely.

### 4.4. Cancer

The tumor microenvironment is associated with increased acidity and hypoxia, decreased nutrients, and increased misfolded proteins. Each of these environmental stressors is a putative trigger for the UPR, and UPR activation is documented in many cancer types, including breast, prostate, colorectal, and glioma [72,73,74]. Colorectal cancer, gliomas, and melanoma have also been linked with increased KDEL receptor expression (Table 1 [56,58,59,60]). In particular, KDELR2 upregulation has been associated with poor outcomes, perhaps because of the role KDELR2 may play in promoting cell proliferation through the mTORC1 pathway [61,75]. Furthermore, tumorigenic cells have an increased demand for protein synthesis in order to support tumor proliferation, migration, and differentiation. An increase in KDEL receptors could benefit tumor cells by ensuring necessary ER resident proteins are maintained within the ER lumen to assist with protein folding, modification, and trafficking. In the case of cancer, KDEL receptor upregulation, while perhaps beneficial for cell survival, may be overall detrimental in terms of disease progression. In cancer, activation of the IRE1 pathway promotes tumor growth, allowing cells to continue growing even as they are confronted by cellular stressors like oncogene expression and an unfavorable microenvironment. IRE1 inhibitors have been used alone or in combination with other drugs to demonstrate that the modulation of the IRE1 pathway has beneficial effects in various cancer models [76]. If KDEL receptors are activated by XBP1 in cancer cells, the inhibition of the IRE1 pathway would predictably prevent KDEL receptor upregulation and deny tumor cells the benefits afforded them by increased KDEL receptors.

### 4.5. Diabetes

Pancreatic cells rely on the UPR to expand their capacity for protein folding and secretion. The IRE1 pathway is implicated in insulin resistance, diabetes, and obesity, and can modulate glucose and lipid metabolism, insulin signaling, and glucagon secretion [77,78]. KDEL receptors are IRE1/XBP1-regulated genes that are increased in diabetes [63,64,65]. KDEL receptors are also known to interact with prohibitin-1, a multi-functional protein with multiple cellular locales (i.e., nucleus and mitochondria) that is expressed in pancreatic β-cells and may be involved in diabetes [79,80]. KDEL receptors form complexes with prohibitin-1 and prohibitin-1 assists in the retention of KDEL receptors in the cis-Golgi to allow for Src family kinase activation [81]. Activated Src family kinases, in turn, play a role in EGFR transactivation, MAPK activation, and collagen IV synthesis [82]. Increases in KDEL receptors may increase Src kinase activation, which is thought to be a characteristic of diabetes, via its involvement in multiple signaling pathways that are activated in diabetes.

### 4.6. Immune Response

Immune cells are highly secretory cells that exhibit both a physiological and a pathological UPR response. For example, XBP1 is essential for physiological B cell differentiation, but also plays a role in ER expansion and increased protein synthesis in pathological conditions [83,84]. KDEL receptors are implicated in immune cell responses with KDELR1 regulating T-cell homeostasis via a role in the integrated stress response. Specifically, mutated KDELR1 is associated with a reduction in T-cell numbers because of an increase in the integrated stress response [85]. This occurs because the mutated KDELR1 lacks an interaction with a phosphatase in the signaling cascade regulating T-cell homeostasis [86]. KDELR1 mutations are also associated with antiviral immunity, with one mutation (Y158C) being associated with lymphopenia, reduced T-cell receptor expression, deficient antiviral immunity, and increased CD44 expression [87]. While KDELR1 is demonstrated to have a role in immune responses, the function of KDELR2 and KDELR3 remains to be determined. Further examination of the isoform specific responses of the KDEL receptors will help to parse out the redundant and unique functions of the receptors in immunity.

### 4.7. Osteogenesis Imperfecta

Osteogenesis imperfecta (OI; OMIM #166200) is a group of rare connective tissue disorders primarily characterized by an increased propensity for bone fractures. Secondary symptoms include progressive hearing loss, blue sclera, joint hypermobility, and dentinogenesis imperfecta [88]. Recently, bi-allelic pathogenic variants in *KDELR2* have been reported in four families with OI. These variants cause KDELR2 protein to insufficiently interact with the ER cargo and display impaired retrograde transport to the ER [88]. Intracellular levels of ERS proteins, Serpin H1 (*HSP47*), and FKBP65 (*FKBP10*), were shown to be decreased in patient-derived fibroblasts, with increased extracellular Serpin H1, supporting improper KDELR2-mediated transport [88]. Elevated extracellular levels of Serpin H1 can hinder proper fiber formation because of its ability to bind to collagen, and likely contribute to the pathogenesis of OI [88]. Another study using OI patient-derived fibroblasts demonstrated UPR activation, evidenced by increased levels of BiP, PDI, and phosphorylated PERK [89]. Interestingly, no difference in spliced XBP1 was detected between the controls and OI fibroblasts. The authors note the limitations of their ex vivo design, suggesting the need for in vivo models to fully decipher the role of UPR activation in OI. Examining KDEL receptor isoform expression from OI fibroblasts may corroborate UPR activation.

### 4.8. Wolfram’s Syndrome

Wolfram syndrome is a rare, autosomal recessive disease characterized by early onset diabetes, progressive optic atrophy, and deafness. Similar to diabetes, XBP1 signaling is implicated in Wolfram syndrome [90]. As early onset diabetes is a hallmark of Wolfram syndrome, it is highly likely that KDEL receptors are similarly activated through the XBP1 pathway in this disease. Wolfram syndrome is caused by mutations in two genes: *WFS1* and *CISD2. WFS1* codes for the protein Wolframin, which is an ER-associated transmembrane glycoprotein thought to play a role in regulating ER calcium homeostasis [91]. One study showed that therapeutically relevant mutations in *WFS1* caused increased ER calcium dysregulation and the induction of UPR [92]. As calcium dysregulation is known to increase KDEL receptor expression, in conjunction with the knowledge that the IRE1 pathway is activated, it is probable KDEL receptors are considerably impacted by this disease and represents an area of important future investigation [8].

## 5. Discussion

KDEL receptors mediate the retrieval of ERS proteins from the Golgi to the ER. Many proteins containing an ERS are essential for nascent protein maturation in the ER (e.g., chaperones and isomerases) and some of these same ERS proteins are UPR genes (Table 2). KDEL receptor expression is increased during the UPR through activation of the IRE1 pathway. Under ER stress conditions where ER calcium decreases, ERS-containing proteins are secreted en masse from the cell. The upregulation of KDEL receptors during UPR activation may serve to prevent the loss of existing ERS proteins, as well as the newly made UPR genes as the KDEL receptors work to stabilize the ER proteome and restore cellular proteostasis (Figure 1).

In addition to their role in the retrieval of ERS-containing proteins from the Golgi to the ER, KDELR receptors can affect membrane trafficking to cause lysosomal rearrangement and promote autophagy [35,36]. KDEL receptors can also activate Src kinase to promote membrane trafficking and the degradation of the extracellular matrix [33,93]. Given their role in trafficking and autophagy, KDEL receptors may also promote the degradation and removal of misfolded proteins during ER stress. KDEL receptors traffic to the cell surface basally and under stress conditions, where they can interact with ERS ligands to form clusters and potentially promote signaling [21,23,24,94,95]. The presence of KDEL receptors at the cell surface may function to retrieve secreted ERS proteins from the extracellular space. Additionally, KDEL receptors on the cell surface may detect elevated ERS proteins in the extracellular space, and promote intracellular signaling (e.g., a paracrine sensor of UPR via ERS proteins acting as “alarmones”). KDEL receptors have also recently been shown to modulate the trafficking of sodium channels to the surface, indicating a more general role in the maturation of transmembrane proteins destined for the cell surface [96]. The upregulation of KDEL receptors during the UPR may be critical for UPR functions beyond ERS protein retrieval from the Golgi to ER. Additional studies into the functions of KDEL receptors during a UPR may provide insight into their role in diseases associated with ER stress and UPR.

The temporal changes in KDEL receptors in response to pathological states needs be further studied to understand their role as UPR genes. Techniques used to examine KDEL receptor expression (i.e., RNAseq and qPCR) provide information about KDEL receptor expression at a single timepoint and may not capture the dynamic nature of these receptors. For example, *KDELR3* is upregulated 24h after oxygen–glucose deprivation, but not immediately following or 8h after the insult [8]. The timepoints studied may account for whether the upregulation or downregulation of KDEL receptors is observed in different pathological conditions (Table 1). We have discussed how KDEL receptors are upregulated UPR genes in response to ER stressors, but KDEL receptor downregulation is also of interest. For example, in cancer models, the loss of KDEL receptors may be beneficial in preventing cancer by hindering cell proliferation. In other situations, such as the rejection of a kidney transplant, it was noted that the *KDELR1* expression was decreased, but no causative relationship was examined [71]. Future studies using CRISPRa and CRISPRi approaches to regulate the expression of endogenous KDEL receptors in different disease models may identify new therapeutic opportunities for treating diseases [97].

The regulation of KDEL receptors using gene therapy may ultimately be a viable treatment for a variety of diseases. For example, under conditions of ER calcium depletion, ERS proteins localized to the ER lumen are secreted from the cell [8]. Overexpressing KDEL receptors have been shown to attenuate the loss of ERS proteins in an oxygen–glucose deprivation model, suggesting that KDEL receptor augmentation may be useful in restoring proteostasis following an ischemic injury [8]. Similarly, providing exogenous KDEL receptors via gene delivery may be a viable option to improve outcomes in diseases where the disruption of ER proteostasis has been identified. In addition, known mutations in *KDELR1* and *KDELR2* are associated with lymphopenia and osteogenesis imperfecta, respectively, making KDEL receptor replacement an attractive therapeutic strategy for these conditions [87,88].

Diverse disease states, beyond those explored above, are also associated with changes in KDEL receptor expression (Table 1). While many of these diseases are thought to be linked to UPR activation, concurrent examinations of KDEL receptors and UPR were difficult to find. Given the nature of KDEL receptors as adaptive UPR genes involved in maintaining the ER proteome, further investigation into their role in disease is warranted. We put forth that KDEL receptors are UPR genes [8], and others have suggested that KDEL receptors are regulators of UPR [47,98]. Beyond their canonical function as part of the KDEL retrieval pathway, we describe above how KDEL receptors participate in autophagy, membrane trafficking, intracellular signaling, and retrieval of ER proteins from the extracellular space. These ascribed functions of the KDEL receptor go beyond the retrieval of the ERS proteins from the Golgi to the ER and should be examined as part of the UPR under physiological and pathological conditions (Figure 1).

## Figures and Tables

**Figure 1 ijms-22-05436-f001:**
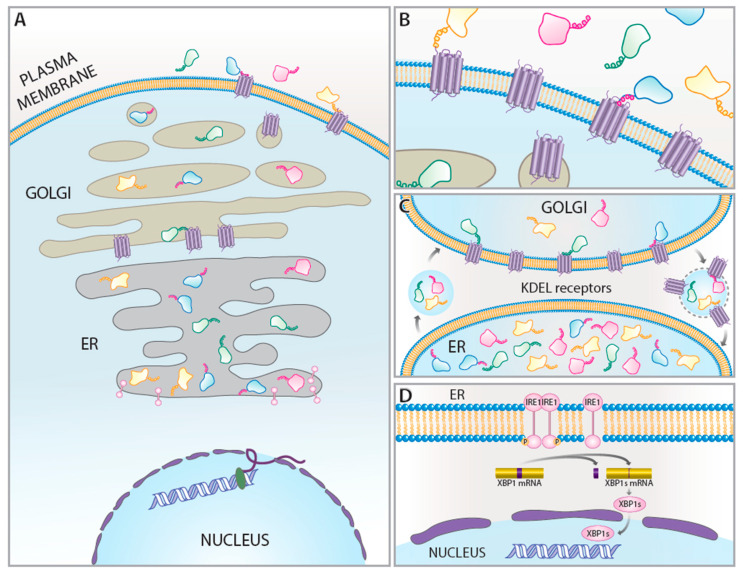
(**A**) When proteins with ER retention sequences move along the secretory pathway they encounter KDEL receptors in the *cis*-Golgi. KDEL rectors recognize the carboxy terminus ER retention sequence of these proteins and traffic the proteins back to the ER lumen. (**B**) KDEL receptors in the plasma membrane modulate cell surface binding of ERS-containing proteins. Following ER stress, there is an increase in KDEL receptors found at the cell surface. (**C**) ERS-containing proteins that escape from the ER interact with KDEL receptors in the Golgi. The KDEL receptor-protein complex moves from the Golgi to the ER through COPI-mediated retrograde transport and the ER protein is release into the ER lumen. (**D**) ER stress leads to dissociation of BiP and IRE1α complexes, allowing for IRE1α oligomerization. Auto-phosphorylation of oligomerized IRE1α activates its kinase and RNase activities to initiate *XBP1* splicing. Spliced *XBP1* translocates to the nucleus to induce transcription of UPR genes, like the KDEL receptors.

**Table 1 ijms-22-05436-t001:** *KDEL* receptor expression is altered in disease states. Arrows indicate changes in gene expression.

Model	*KDELR1*	*KDELR2*	*KDELR3*	Reference
Ischemia				
Cerebral ischemic stroke RNAseq	↑	↑	↑	[52]
Myocardial infarction RNAseq	↑	↑		[53]
Myocardial infarction with ischemic postconditioning RNAseq	↓	↓		[53]
SH-SY5Y oxygen–glucose deprivation RT-qPCR			↑	[8]
Ischemic cardiomyopathy RNAseq	↑			[54]
Cancer				
Colorectal carcinoma CDNA microarray			↑	[56]
YCC-16 gastric cancer cell line cDNA microarray			↓	[57]
Glioma tissue TCGA, CGGA and GSE16011 database		↑		[58]
Melanoma human tumor RNAseq			↑	[59]
FMPC optic glioma RNAseq			↑	[60]
Glioblastoma tissue qRT-PCR		↑		[61]
Pancreatic Disorders				
Pancreatic islets modified by palmitate RNAseq	↓			[62]
Type 2 Diabetes GeneChip expression array		↑	↑	[63]
Type 2 Diabetes db/db mice RNAseq			↑	[64]
Diabetic glomeruli and tubulointerstitium RNAseq		↑	↑	[65]
Heart Disease				
Dilated cardiomyopathy RNAseq		↓		[54]
Heart failure RNAseq	↓	↑		[66]
Dilated cardiomyopathy RNAseq	↑			[67]
Other				
Liver regeneration following partial hepatectomy RNAseq			↑	[68]
Liver regeneration following portal vein ligation RNAseq			↑	[68]
Liver regeneration following associated liver partition and portal vein ligation for staged hepatectomy RNAseq			↑	[68]
Chronic obstructive pulmonary disease RNAseq			↑	[69]
Irritable bowel syndrome rectal biopsy qPCR		↑		[70]
Subclinical acute rejection of kidney transplant RNAseq	↓			[71]

**Table 2 ijms-22-05436-t002:** ERS-containing proteins are UPR regulated genes.

Protein	Symbol	C-Terminus	Function
BiP	HSPA5	TAEKDEL	Chaperone
Grp94	HSP90B1	TAEKDEL	Chaperone
Calreticulin	CALR	GQAKDEL	Chaperone
Protein disulfide isomerase	P4HB	KAVKDEL	Isomerase
Protein disulfide isomerase A3	PDIA3	KKAQEDL	Isomerase
Protein disulfide isomerase A4	PDIA4	SRTKEEL	Isomerase
Protein disulfide isomerase A5	PDIA5	GKKKEEL	Isomerase
Protein disulfide isomerase A6	PDIA6	DLGKDEL	Isomerase
Hypoxia up-regulated protein 1	HYOU1	PLKNDEL	Chaperone
Mesencephalic astrocyte-derived neurotrophic factor	MANF	ASARTDL	Pleiotropic
Cerebral dopamine neurotrophic factor	CDNF	THPKTEL	Pleiotropic

## Data Availability

Not applicable.

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
