# Peer review of "The Function of KDEL Receptors as UPR Genes in Disease"

_ijms, 2021, doi:10.3390/ijms22115436_

Round 1
Reviewer 1 Report
In this review the authors discuss the role of KDEL receptors as unfolded protein response (UPR) genes involved in diseases. They first mention their canonical role in retrieving proteins from the Golgi to the ER before describing their functions as actors of the UPR. Thereby different aspects on how KDEL receptors may act in response to ER stress to restore proteostasis are detailed, including retrieval of ER proteins from the extracellular space, roles in autophagy and membrane trafficking. In a second part, the authors refer to the information given in the first part to discuss various findings on the expression and involvement of KDEL receptors in neurodegeneration, cancer, diabetes, ischemia, immune response, osteogenis imperfecta and Wolfram’s syndrome. Finally they discuss the fact that, although many points still need to be investigated, the regulation of KDEL receptor expression may be an option in the therapy of some of these diseases.
The review is very well written, data gathered from numerous sources, synthetized in a way that gives a very good overall view of the topic and thoroughly discussed. The tables and the figure support the text and allow to get important facts at a glance. The information given gets to the point, keeping to the topic of KDEL receptors as UPR genes without going into too much details and extended bibliography is provided for further reading. Thus this review is very informative, interesting and, additionally, also pleasant to read.
Author Response
We thank the reviewer for his/her favorable review of our manuscript.
Reviewer 2 Report
This is a very interested review about the role of the KDEL receptors as UPR target genes. I feel that the review summarizes new advances in the field and should be of interest for the readership of IJMS. I only have minor comments.
The authors should be careful of citing original literature. An example of that is : "The IRE1 pathway was originally identified in yeast and is the most evolutionarily conserved arm of the UPR [26]". the citation is review. They should cite the original work from the Walter lab.
In the paragraph entitled Neurodegeneration, only 2 references are cited. while I understand this is not the central point of the review, it should tell the reader that a lot more is known about UPR and neurogegeneration. Something like, it is summarized in this review, her are some specific examples.
Other than that, I enjoyed the manuscript. Good work .
Author Response
We thank the reviewer for his/her comments. We added the Walter lab citation to include the original work pertaining to IRE1 (see line 123, reference 26). Additionally, we edited the neurodegeneration section to include references regarding UPR and neurodegeneration (see lines 126-128, references 41 & 42).